# PyTrial: Machine Learning Software and Benchmark for Clinical Trial Applications

## Abstract

Clinical trials are conducted to test the effectiveness and safety of potential drugs in humans for regulatory approval. Machine learning (ML) has recently emerged as a new tool to assist in clinical trials. Despite this progress, there have been few efforts to document and benchmark ML4Trial algorithms available to the ML research community. Additionally, the accessibility to clinical trial-related datasets is limited, and there is a lack of well-defined clinical tasks to facilitate the development of new algorithms.

To fill this gap, we have developed `PyTrial` that provides benchmarks and open-source implementations of a series of ML algorithms for clinical trial design and operations. In this paper, we thoroughly investigate 34 ML algorithms for clinical trials across 6 different tasks, including patient outcome prediction, trial site selection, trial outcome prediction, patient-trial matching, trial similarity search, and synthetic data generation. We have also collected and prepared 23 ML-ready datasets as well as their working examples in Jupyter Notebooks for quick implementation and testing.

`PyTrial` defines each task through a simple four-step process: data loading, model specification, model training, and model evaluation, all achievable with just a few lines of code. Furthermore, our modular API architecture empowers practitioners to expand the framework to incorporate new algorithms and tasks effortlessly.

## 1 Introduction

Developing a novel drug molecule from its initial concept to reaching the market typically involves a lengthy process lasting between 7 to 11 years and an average cost of \$2 billion (Martin et al., 2017). Drug development has two major steps: discovery and clinical trials. Discovery aims to find novel drug molecules with desirable properties, while drug development through clinical trials assesses their safety and effectiveness. A new drug must pass through phases I, II, and III of clinical trials to be approved by the FDA. Phase IV trials are conducted after approval to monitor the drug's safety and effectiveness. These stages require significant time, investment, and resources.

Machine learning (ML) methods offer a promising avenue to reduce costs and accelerate the drug development process. Over the past few years, there has been an increasing number of works published in the field of ML for drug discovery (Du et al., 2022; Jin et al., 2018; Nigam et al., 2020; Brown et al., 2019; Fu et al., 2021a; 2022a) and development (Wang et al., 2022; Wang & Sun, 2022c;b; Zhang et al., 2020; Gao et al., 2020a; Fu et al., 2021b; Wang & Sun, 2022a; Wang et al., 2023a). Although there were efforts in developing benchmarking platforms (Huang et al., 2021b; Gao et al., 2022) and software solutions (Zhu et al., 2022; Brown et al., 2019) for ML for drug discovery methods, the field of ML4Trial has not seen the same level of systematic development and documentation (Wang et al., 2022). This can be attributed to the absence of benchmark works and the lack of clear definitions to formulate clinical trial problems as ML tasks.

This paper presents a comprehensive benchmark called `PyTrial` that aggregates the mainstream ML methods for clinical trial tasks. The overview is shown in Figure 1. `PyTrial` involves 6 ML4Trial tasks, including *Patient Outcome Prediction*, *Patient-Trial Matching*, *Trial Site Selection*, *Trial Search*, *Trial Outcome Prediction*, and *Patient Data Simulation*. We conclude the 4 data ingredients for these tasks by *Patient*, *Trial*, *Drug*, and *Disease*, hence defining a unified data loading API.

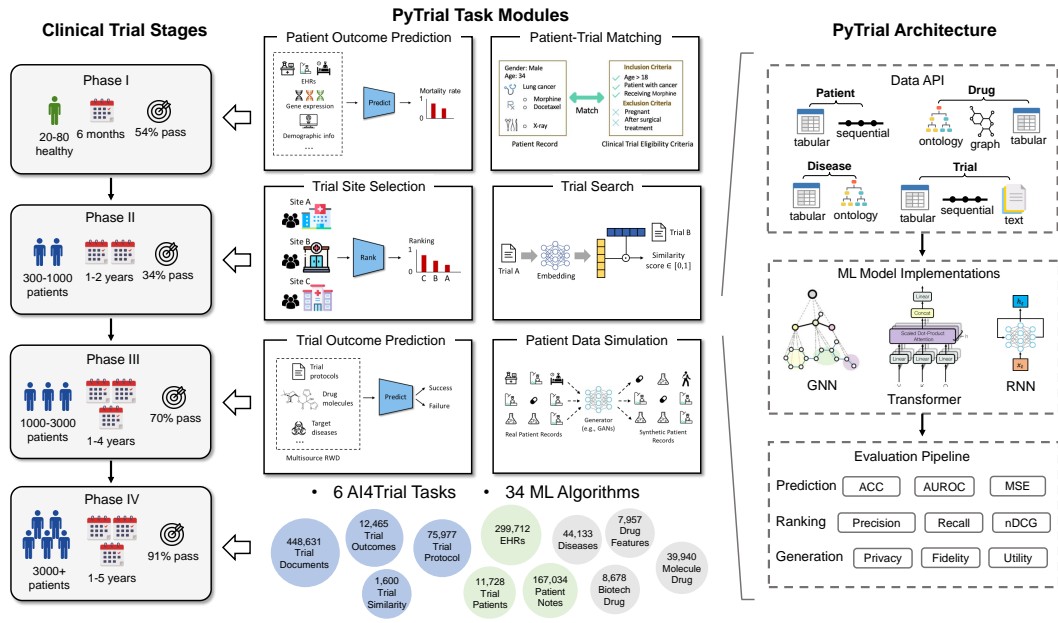

Figure 1: The `PyTrial` platform combines a comprehensive set of AI tools for clinical trial tasks with over 30 implemented machine learning algorithms. Designed as a Python package, `PyTrial` provides practitioners with a versatile solution to harness AI capabilities throughout all phases of clinical trials. It is featured for a unified data API encompassing patients, drugs, diseases, and trials, as well as user-friendly ML algorithms and a standardized evaluation pipeline.

Correspondingly, we offer more than 20 ML-ready datasets for fast verification and development of ML models. At last, we develop a standard evaluation pipeline for all tasks, such as *accuracy* for prediction tasks, *precision/recall* for ranking tasks, and *privacy*, *fidelity*, and *utility* for generation tasks. For all algorithms involved in `PyTrial`, we provide a working example in Jupyter Notebook to ensure convenient testing and implementation.

The software `PyTrial` provides the following contribution related to ML for clinical trials:

- **Problems:** We have systematically formulated ML tasks for clinical trial applications and presented them in a concise summary.
- **Algorithms:** Our study includes a comprehensive evaluation of over 30 AI methods across 6 mainstream AI algorithms for clinical trial problems.
- **API:** We offer a unified API for data loading, model training, and model deployment, with interactive examples, making it easy for users to implement ML algorithms for clinical trials with just a few lines of code.
- **Datasets:** Our study provides 23 datasets covering patients, trials, diseases, and drugs, which are readily available for use in drug development through ML algorithms.

In a nutshell, `PyTrial` offers a comprehensive interface to support the rapid implementation of ML4Trial algorithms on users' own data and the deployment of these algorithms to enhance clinical trial planning and running. It also enables future ML4Trial research by providing a well-defined new benchmark.

## 2 SOFTWARE DESIGN AND IMPLEMENTATION

### 2.1 SOFTWARE STRUCTURE

The overall structure of `PyTrial` is in Figure 1. We create a hierarchical framework that comprises three primary layers: (1) unified data API, (2) task modules for AI models, and (3) prediction and evaluation pipeline. We also maintain a standardized ML pipeline for executing all tasks and models.

Within `PyTrial`, tasks are defined based on their input and output data, which can be quickly loaded via the API as

```python
"""An example of building sequential patient data for patient outcome
    prediction."""
# load demo data
from pytrial.data.demo_data import load_synthetic_ehr_sequence
data = load_synthetic_ehr_sequence()

# prepare input for the model
from pytrial.tasks.indiv_outcome.data import SequencePatient
data = SequencePatient(data={
        "v":data["visit"], # sequence of visits
        "y":data["y"], # target labels to predict
        "x":data["feature"]}, # static baseline features
    metadata={"voc":data["voc"] # vocabulary for events
    })
```

Once we specify the training data, `PyTrial` offers a standard workflow `load data` → `model definition` → `model training` → `model evaluation` as

```python
"""An example of training and testing patient outcome prediction
    model."""
# init model
from pytrial.tasks.indiv_outcome.sequence import RNN
model = RNN()

# fit model
model.fit(data)

# make predictions
model.predict(data)

# save model
model.save_model("./checkpoints")
```

It is important to highlight that we maintain a consistent model API for all tasks, ensuring a seamless transition when users adopt a new model or engage in a different task. This approach mitigates the gaps or inconsistencies in the user experience.

## 2.2 PYTRIAL DATA MODULES

We categorize the modalities of input data for clinical trial tasks by *patient*, *trial*, *drug*, and *disease*. Users can create the inputs for the task modules by composing these data modules. A series of pre-processed datasets are also provided for quick adoption of ML algorithms, as shown in Table 1.

**Patient Data**. We classify patient data by *tabular* and *sequential* datasets. Tabular patient data represents the static patient features stored in a spreadsheet, i.e., one patient data $\mathbf{x} = \{x_1, x_2, \dots\}$ where each $x_*$ is a binary, categorical, or numerical feature. Sequential data represents multiple admissions of a patient that are in chronological order, as $\mathbf{V}_{1:T} = \{\mathbf{V}_1, \mathbf{V}_2, \dots, \mathbf{V}_T\}$, where an admission $\mathbf{V}_* = \{\mathbf{v}_1, \mathbf{v}_2, \dots\}$ constitutes a bunch of events $\mathbf{v}_*$ occurred at the same time.

**Trial Data**. We refer clinical trial data to the trial protocols written in lengthy documents[1]. Considering clinical trial documents' meta-structure, we can extract key information and reorganize the trial data into a tabular format, represented as $\mathbf{t} = \{t_1, t_2, \dots\}$. Each element $t_*$ in this data structure can correspond to a section or a feature of the clinical trial. $\mathbf{t}$ can be utilized for diverse tasks such as trial outcome prediction and trial design. Furthermore, considering the topic similarity and timestamp of trials, we can reformulate tabular trial data as sequences, i.e., a trial topic

---

[1] https://ClinicalTrials.gov

Table 1: The list of ML4Trial datasets integrated into the `PyTrial` platform.

| Dataset Name | Sample Size | Data Format | Source |
|---|---|---|---|
| Patient: NCT00041119 (Wang & Sun, 2022b) | 3,871 | Patient - Tabular | PDS |
| Patient: NCT00174655 (Wang & Sun, 2022b) | 994 | Patient - Tabular | PDS |
| Patient: NCT00312208 (Wang & Sun, 2022b) | 1,651 | Patient - Tabular | PDS |
| Patient: NCT00079274 (Wang & Sun, 2022b) | 2,968 | Patient - Tabular | PDS |
| Patient: NCT00003299 (Wang et al., 2023a) | 587 | Patient - Tabular | PDS |
| Patient: NCT00694382 (Wang & Sun, 2022b) | 1,604 | Patient - Tabular | PDS |
| Patient: NCT03041311 (Wang et al., 2023a) | 53 | Patient - Tabular | PDS |
| PMC-Patient Notes (Zhao et al., 2023) | 167,034 | Patient - Tabular | PubMed |
| Patient: NCT00694382 (Das et al., 2023) | 971 | Patient - Sequential | PDS |
| Patient: NCT01439568 (Das et al., 2023) | 77 | Patient - Sequential | PDS |
| MIMIC-III EHR (Johnson et al., 2016) | 38,597 | Patient - Sequential | MIMIC |
| MIMIC-IV EHR (Johnson et al., 2023) | 143,019 | Patient - Sequential | MIMIC |
| Patient Matching Collection (Koopman & Zuccon, 2016) | 4,000 | Patient - Tabular, Trial - Text | SIGIR |
| TOP Phase I (Fu et al., 2022b; Wang et al., 2023c) | 1,787 | Trial - Tabular, Trial - Sequential | ClinicalTrials.gov |
| TOP Phase II (Fu et al., 2022b; Wang et al., 2023c) | 6,102 | Trial - Tabular, Trial - Sequential | ClinicalTrials.gov |
| TOP Phase III (Fu et al., 2022b; Wang et al., 2023c) | 4,576 | Trial - Tabular, Trial - Sequential | ClinicalTrials.gov |
| Trial Termination Prediction (Wang et al., 2023a) | 223,613 | Trial - Tabular | ClinicalTrials.gov |
| Trial Similarity (Wang & Sun, 2022c) | 1,600 | Trial - Text | ClinicalTrials.gov |
| Eligibility Criteria Design (Wang et al., 2023b) | 75,977 | Trial - Text | ClinicalTrials.gov |
| Clinical Trial Documents (Wang & Sun, 2022c) | 447,709 | Trial - Text | ClinicalTrials.gov |
| Diseases (Chandak et al., 2023) | 17,080 | Disease - Tabular, Disease - Ontology | PrimeKG |
| Drug SMILES (Wishart et al., 2006) | 6,948 | Drug - Graph | Drugbank |
| Drug Features (Wishart et al., 2006) | 7,957 | Drug - Tabular | Drugbank |
| Drug ATC Codes | 6,765 | Drug - Ontology | WHO |

$\mathbf{T}_{1:T} = \{\mathbf{T}_1, \mathbf{T}_2, \ldots, \mathbf{T}_T\}$, where each $\mathbf{T}_* = \{\mathbf{t}_1, \mathbf{t}_2, \ldots\}$ contains a set of trials $\mathbf{t}_*$ started concurrently.

**Drug Data**. The structure of small molecule drugs can be described by SMILES strings (Weininger, 1988), which is amenable to graphical deep learning (Kipf & Welling, 2016). We further enrich the drug data with their properties to build tabular data as $\mathbf{d} = \{d_1, d_2, \ldots\}$. Moreover, the drug database can be mapped to the ontology $\mathcal{G}_{\text{drug}} = \{\mathcal{D}, \mathcal{R}\}$, where $\mathcal{D}$ is the node set representing drugs and $\mathcal{R}$ is the edge set, according to the drug's effects on specific organs or systems and its mechanism of action.

**Disease Data**. Disease features are tabular data that can be mapped to standard coding systems, e.g., ICD-10 (Cartwright, 2013), to formulate disease ontology data. Similar to drug ontology, disease ontology can be represented by $\mathcal{G}_{\text{disease}}$ consisting of nodes of diseases.

## 2.3 PYTRIAL TASK MODULES

In this section, we briefly describe the clinical trial task modules. A complete list of tasks and algorithms in `PyTrial` is shown in Table 2.

**Patient Outcome Prediction**. Patient outcome prediction refers to the task of predicting the clinical outcome of individual patients. This is extremely beneficial for clinical trials, as it helps in multiple aspects, such as developing personalized treatment plans to reduce the risk of subjecting patients to ineffective or harmful interventions, improving trial enrollment, and selecting interventions with higher probabilities of success to enhance trial design. For instance, if a patient is predicted to have a high risk of developing adverse outcomes with the new drug, it is safer and more ethical to not recruit this patient for the clinical trial. However, the eligibility criteria should be balanced to ensure broader and representative enrollments while minimizing the risks to individual patients.

*Machine Learning Setup*. The clinical outcome can be either a binary label $y \in \{0, 1\}$, such as mortality, readmission, or continuous values $y \in \mathbb{R}$, like blood pressure or length of stay. The input tabular patient data can be denoted by $\mathbf{x}$, and sequential data can be $\mathbf{V} = \{\mathbf{V}_0, \mathbf{V}_{1:T}\}$, where $\mathbf{V}_0$ and $\mathbf{V}_{1:T}$ are the patient's baseline features and longitudinal records, respectively. The goal of this task is to train an encoder function $g(\cdot)$ that combines and transforms the input $\mathbf{V}$ into a lower-dimensional representation $\mathbf{h}$. Subsequently, a prediction model $f(\cdot)$ is utilized to forecast the target outcome, i.e., $\hat{y} = f(\mathbf{h})$. We implement many patient outcome prediction algorithms for tabular inputs (Gorishniy et al., 2021; Wang & Sun, 2022b; Wang et al., 2023a) and sequential inputs (Choi et al., 2016a;b; Xu et al., 2018; Ma et al., 2017b; Gao et al., 2020b).

Table 2: The list of ML4Trial algorithms implemented in the `PyTrial` platform.

| Task | Method | Input Data | Module |
|---|---|---|---|
| Patient Outcome Prediction | Logistic Regression (Wang & Sun, 2022b) | Patient - Tabular | `indiv_outcome.tabular.LogisticRegression` |
| | XGBoost (Wang & Sun, 2022b) | Patient - Tabular | `indiv_outcome.tabular.XGBoost` |
| | MLP (Wang & Sun, 2022b) | Patient - Tabular | `indiv_outcome.tabular.MLP` |
| | FT-Transformer (Gorishniy et al., 2021) | Patient - Tabular | `indiv_outcome.tabular.FTTransformer` |
| | TransTab (Wang & Sun, 2022b) | Patient - Tabular | `indiv_outcome.tabular.TransTab` |
| | AnyPredict (Wang et al., 2023a) | Patient - Tabular | `indiv_outcome.tabular.AnyPredict` |
| | RNN (Choi et al., 2016a) | Patient - Sequential | `indiv_outcome.sequence.RNN` |
| | RETAIN (Choi et al., 2016b) | Patient - Sequential | `indiv_outcome.sequence.RETAIN` |
| | RAIM (Xu et al., 2018) | Patient - Sequential | `indiv_outcome.sequence.RAIM` |
| | Dipole (Ma et al., 2017a) | Patient - Sequential | `indiv_outcome.sequence.Dipole` |
| | StageNet (Gao et al., 2020b) | Patient - Sequential | `indiv_outcome.sequence.StageNet` |
| Trial Site Selection | PG-Entropy (Srinivasa et al., 2022) | Trial - Tabular | `site_selection.PolicyGradientEntropy` |
| | FRAMM (Theodorou et al., 2023a) | Trial - Tabular | `site_selection.FRAMM` |
| Trial Outcome Prediction | Logistic Regression (Fu et al., 2022b) | Trial - Tabular | `trial_outcome.LogisticRegression` |
| | MLP (Fu et al., 2022b) | Trial - Tabular | `trial_outcome.MLP` |
| | XGBoost (Fu et al., 2022b) | Trial - Tabular | `trial_outcome.XGBoost` |
| | HINT (Fu et al., 2022b) | Trial - Tabular | `trial_outcome.HINT` |
| | SPOT (Wang et al., 2023c) | Trial - Sequential | `trial_outcome.SPOT` |
| | AnyPredict (Wang et al., 2023a) | Trial - Tabular | `trial_outcome.AnyPredict` |
| Patient-Trial Matching | DeepEnroll (Zhang et al., 2020) | Trial - Text, Patient - Sequential | `trial_patient_match.DeepEnroll` |
| | COMPOSE (Gao et al., 2020a) | Trial - Text, Patient - Sequential | `trial_patient_match.COMPOSE` |
| Trial Search | BM25 (Wang & Sun, 2022c) | Trial - Text | `trial_search.BM25` |
| | Doc2Vec (Le & Mikolov, 2014) | Trial - Text | `trial_search.Doc2Vec` |
| | WhitenBERT (Huang et al., 2021a) | Trial - Text | `trial_search.WhitenBERT` |
| | Trial2Vec (Wang & Sun, 2022c) | Trial - Text | `trial_search.Trial2Vec` |
| Trial Patient Simulation | GaussianCopula (Sun et al., 2019) | Patient - Tabular | `trial_simulation.tabular.GaussianCopula` |
| | CopulaGAN (Sun et al., 2019) | Patient - Tabular | `trial_simulation.tabular.CopulaGAN` |
| | TVAE (Xu et al., 2019) | Patient - Tabular | `trial_simulation.tabular.TVAE` |
| | CTGAN (Xu et al., 2019) | Patient - Tabular | `trial_simulation.tabular.CTGAN` |
| | MedGAN (Choi et al., 2017) | Patient - Tabular | `trial_simulation.tabular.MedGAN` |
| | RNNGAN (Wang & Sun, 2022a) | Patient - Sequential | `trial_simulation.sequence.RNNGAN` |
| | EVA (Biswal et al., 2021) | Patient - Sequential | `trial_simulation.sequence.EVA` |
| | SynTEG (Zhang et al., 2021) | Patient - Sequential | `trial_simulation.sequence.SynTEG` |
| | PromptEHR (Wang & Sun, 2022a) | Patient - Sequential | `trial_simulation.sequence.PromptEHR` |
| | Simulants (Beigi et al., 2022) | Patient - Sequential | `trial_simulation.sequence.KNNSampler` |
| | TWIN (Das et al., 2023) | Patient - Sequential | `trial_simulation.sequence.TWIN` |

**Trial Site Selection**. Effective clinical trial operation depends on identifying the best clinical sites and investigators. To achieve this, we need to recruit those sites and investigators that possess the clinical expertise and patient demographics required for the trial. Balancing patient enrollment number, patient diversity, and quality/cost of the site is critical to ensure optimal results.

When initiating a new clinical trial, Contract Research Organizations (CROs) select investigators from a large pool using a set of predefined criteria. The task of matching a trial site is posed as a fair ranking problem, where the list of potential trial sites is ranked to maximize patient enrollment and diversity. Algorithms used in this process optimize enrollment by evaluating investigator performance records and patient demographics, thereby refining the selection process. They also promote diversity by facilitating the inclusion of underrepresented populations in clinical trials, aligning with regulatory recommendations and fostering comprehensive research outcomes.

*Machine Learning Setup.* Trial site selection is a crucial aspect of clinical trials, aiming to identify the most suitable sites from the candidate set $\mathbf{S} = \{\mathbf{s}_1, \mathbf{s}_2, \dots\}$, for recruiting diverse and sufficiently numbered patients to evaluate the treatment's effectiveness and safety. It is framed as a ranking problem, generating a ranking $\mathcal{R}$ over $\mathbf{S}$ based on the trial $\mathbf{t}$ in order to select a subset of the highest-ranked sites. The goal is then to learn a policy $\pi$ mapping $\mathbf{t}$ to a ranking (or distribution of rankings) such that we minimize $\ell(\pi; \mathbf{S}, \mathbf{t})$, a predefined loss function measuring enrollment, diversity, and/or any other factor over the subset of sites selected (as measured by being ranked above some threshold). We incorporate Policy gradient entropy (PGentropy) (Srinivasa et al., 2022) and Fair Ranking with Missing Modalities (FRAMM) (Theodorou et al., 2023a) for this problem.

**Trial Outcome Prediction**. Assessing patient-level outcomes is essential, but predicting trial-level outcomes accurately is equally important (Fu et al., 2022b; Wang et al., 2023c). It helps in clinical trial planning and saves resources and time by avoiding high-risk trials. Note that it is important to balance this outcome risk assessment with ethical considerations that take into account the fairness, value, and importance of the trial. The main objective for trial outcome prediction is to evaluate the likelihood of a trial's success based on diverse information such as the target disease, drug candidate, eligibility criteria for patient recruitment, and other trial design considerations. Algorithms can optimize the trial parameters according to AI predictions, which improves the trial design and reduces the likelihood of inconclusive or failed results due to poorly designed eligibility criteria, outcome measures, or experimental arms.

*Machine Learning Setup*. This task is framed as a prediction problem where the target $y \in \{0, 1\}$ is a binary indicator of whether the trial would succeed in getting approved for commercialization. We need to implement an encoder $g(\cdot)$ that encodes multi-modal trial data, e.g., text, table, or sequence, into dense embeddings $\mathbf{h}$. A prediction model $f(\cdot)$ then forecasts the trial outcome $\hat{y} = f(\mathbf{h})$. We incorporate trial outcome prediction algorithms for tabular inputs (Fu et al., 2022b) and for sequential inputs (Wang et al., 2023c).

**Patient-Trial Matching**. Failing to enroll sufficient subjects in a trial is a long-standing problem: more than 60% of trials are delayed due to lacking accrual, which causes potential losses of $600K per day (Ness, 2022). ML is promising to accelerate the patient identification process where it selects the appropriate patients that match the trial eligibility criteria based on their records. It also builds the connection between eager patients and suitable trials to improve overall patient engagement.

*Machine Learning Setup*. Formally, this task is formulated as a ranking problem: given the patient sequential data $\mathbf{V}_{1:T} = \{\mathbf{V}_1, \mathbf{V}_2, \dots\}$ and text trial data $\{\mathbf{I}, \mathbf{E}\}$, where $\mathbf{I} = \{\mathbf{i}_1, \mathbf{i}_2, \dots\}$ are inclusion criteria and $\mathbf{E} = \{\mathbf{e}_1, \mathbf{e}_2, \dots\}$ are exclusion criteria. The target is to minimize the distance of $\mathbf{V}$ and $\{\mathbf{i}\}$ and maximize the distance of $\mathbf{V}_{1:T}$ and $\{\mathbf{e}\}$ if the patient matches the trial. Our package involves DeepEnroll (Zhang et al., 2020) and Cross-Modal Pseudo-Siamese Network (COMPOSE) (Gao et al., 2020a).

**Trial Search**. The task of trial search involves finding relevant clinical trials based on a given query or input trial. It enables the efficient identification of relevant trials during the trial design and planning phase. Trial search facilitates trial design by referring to prior trial protocols and results, which can provide important reference points related to the trial design, such as control arms, outcome measures and endpoints, sample size, eligibility criteria.

*Machine Learning Setup*. This task is formulated as a retrieval problem, where an encoder function $f(\cdot)$ is utilized to convert the input trial text data or tabular trial data $\mathbf{t} = \{t_1, t_2, \dots\}$ (where each $t$ indicates a section of the document) into semantically meaningful embeddings $\mathbf{h}$. We implemented pre-trained language models (Devlin et al., 2019) and self-supervised document embedding methods (Wang & Sun, 2022c).

**Trial Patient Simulation**. Generating synthetic clinical trial patient records can help unlock data sharing across institutes while protecting patient privacy. This is accomplished by developing generative AI models that learn from real patient data and use that knowledge to create new, synthetic patient data. This can be done through unconditional or conditional generation. The resulting personalized patient data simulation helps balance the generated data and create more records for underrepresented populations. This method also has the potential to reduce the need for patient recruitment while still providing insights into the treatment effects between digital twins. Synthetic patient data can be used to augment control arms in external comparator studies, making this method particularly relevant in those applications Theodorou et al. (2023b); D'Amico et al. (2023).

*Machine Learning Setup*. Formally, we denote a patient data by $\mathbf{X} = \{\mathbf{V}_0, \mathbf{V}_{1:T}\}$ and the training set $\mathcal{X}$. A generator $p(\cdot)$ is trained on the real patient records $\mathcal{V}$ so as to generate synthetic records unconditionally, as $\hat{\mathbf{X}} \sim p(\mathbf{X}|\mathcal{X}; Z)$, where $Z$ is a random noise input; or generate conditioned on manually specified features $\mathbf{X}'$, as $\hat{\mathbf{X}} \sim p(\mathbf{X}|\mathcal{X}; \mathbf{X}')$. As mentioned, we can generate two types of patient data: tabular (Sun et al., 2019; Xu et al., 2019; Choi et al., 2017) and sequential (Biswal et al., 2021; Zhang et al., 2021; Wang & Sun, 2022a; Beigi et al., 2022).

## 2.4 PREDICTION AND EVALUATION PIPELINE

`PyTrial` integrates a series of utility functions for evaluation. Users can refer to the task-specific metrics to evaluate the performances of ML4Trial models. More specifically, the 6 tasks listed in Table 2 can be categorized by *prediction*, *ranking*, and *generation*. Below, we briefly describe the metrics for these tasks. Detailed descriptions can be found in Appendix B.

**Prediction**. For classification tasks, `PyTrial` provides accuracy (ACC), area under ROC curve (AUROC), area under precision-recall curve (PR-AUC) for binary and multi-class classification; F1-score, PR-AUC, Jaccard score for multi-label classification. For regression tasks, mean-squared error (MSE) is offered.

**Ranking**. Based on the retrieved top-$k$ candidates, we compute a series of retrieval metrics, including precision@$K$, recall@$K$, and ndCG@$K$ to measure the ranking quality.

**Generation**. The `PyTrial` framework utilizes metrics to evaluate the *privacy*, *fidelity*, and *utility* aspects of generated synthetic data. The privacy metric assesses the level of resilience of the generated synthetic data against privacy adversaries, including membership inference attacks and attribute disclosure attacks. The fidelity metric quantifies the similarity between the synthetic and original real data. Lastly, the utility metric determines the usefulness of the synthetic data when applied to downstream ML tasks.

## 3 BENCHMARK ANALYSIS

In this section, we describe how we benchmark ML4Trial algorithms using `PyTrial` with the discussions of main findings. We will release the experiment code, the documentation of `PyTrial`, and interactive Colab notebook examples.

We benchmark all the six ML4Trial tasks integrated into `PyTrial` (see Section 2.3). We cover the results of trial site selection and patient trial matching in the Appendix due to the page limit. We used the best hyperparameters for all these methods through validation performances, with details discussed in the Appendix. We picked the benchmark datasets for each task that fit the input data modality and format, as listed in Table 1.

### 3.1 PATIENT OUTCOME PREDICTION

Table 3: The benchmarking results for *patient outcome prediction* for tabular patient datasets. Results are AUROC for patient mortality label prediction (binary classification). "-" implies the model is not converging. The best are in bold.

| Dataset | | Method | | | | | |
|---------|-----------|--------------------|---------|--------|----------------|----------|------------|
| Name | Condition | LogisticRegression | XGBoost | MLP | FT-Transformer | TransTab | AnyPredict |
| Patient: NCT00041119 | Breast Cancer | 0.5301 | 0.5526 | 0.6091 | - | 0.6088 | **0.6262** |
| Patient: NCT00174655 | Breast Cancer | 0.6613 | 0.6827 | 0.6269 | **0.8423** | 0.7359 | 0.8038 |
| Patient: NCT00312208 | Breast Cancer | 0.6012 | 0.6489 | 0.7233 | 0.6532 | 0.7100 | **0.7596** |
| Patient: NCT00079274 | Colorectal Cancer | 0.6231 | 0.6711 | 0.6337 | 0.6386 | **0.7096** | 0.7004 |
| Patient: NCT00003299 | Lung Cancer | 0.6180 | - | 0.7465 | - | 0.6499 | **0.8649** |
| Patient: NCT00694382 | Lung Cancer | 0.5897 | 0.6969 | 0.6547 | **0.7197** | 0.5685 | 0.6802 |
| Patient: NCT03041311 | Lung Cancer | 0.6406 | 0.8393 | - | - | 0.6786 | **0.9286** |

We evaluate the patient outcome prediction algorithms on the tabular clinical trial patient datasets released in (Wang et al., 2023a). Given our focus on patient mortality prediction datasets, we have chosen a wide range of tabular prediction models. These models include traditional options like Logistic Regression and XGBoost (Chen & Guestrin, 2016). We also incorporate modern neural network alternatives like MLP and FT-Transformer (Gorishniy et al., 2021). Additionally, we explore the potential of cross-table transfer learning models TransTab (Wang & Sun, 2022b) and AnyPredict (Wang et al., 2023a). Further details about the selected models and the experimental setups can be found in Appendix C.

The result of AUROC is shown in Table 3. Interestingly, we observed that AnyPredict (Wang et al., 2023a) achieved the best performance on four datasets, primarily due to its ability to leverage transfer learning across tables. On the other hand, FT-Transformer (Gorishniy et al., 2021) performed exceptionally well on two datasets but struggled in converging on two other datasets. This finding suggests that sophisticated deep learning methods excel at representation learning but may require larger amounts of data for accurate tabular patient prediction.

### 3.2 TRIAL OUTCOME PREDICTION

We conducted an evaluation of trial outcome prediction algorithms using the TOP benchmark (Fu et al., 2022b). We have involved a suite of traditional machine learning prediction models such as Logistic Regression, MLP, and XGBoost (Chen & Guestrin, 2016) as the baselines. We also incorporate specific trial outcome prediction models HINT (Fu et al., 2022b), SPOT (Wang et al.,

Table 4: The benchmarking results for *trial outcome prediction* for tabular clinical trial outcome datasets. Results are AUROC and PR-AUC for trial outcome labels (binary classification). The best are in bold.

| Method | TOP Phase I | | TOP Phase II | | TOP Phase III | |
|---|---|---|---|---|---|---|
| | AUROC | PR-AUC | AUROC | PR-AUC | AUROC | PR-AUC |
| LogisticRegression | 0.520 | 0.500 | 0.587 | 0.565 | 0.650 | 0.687 |
| MLP | 0.550 | 0.547 | 0.611 | 0.604 | 0.681 | 0.747 |
| XGBoost | 0.518 | 0.513 | 0.600 | 0.586 | 0.667 | 0.697 |
| HINT | 0.576 | 0.567 | 0.645 | 0.629 | 0.723 | 0.811 |
| SPOT | 0.660 | 0.689 | 0.630 | 0.685 | 0.711 | 0.856 |
| AnyPredict | **0.699** | **0.726** | **0.706** | **0.733** | **0.734** | **0.881** |

2023c), and AnyPredict (Wang et al., 2023a). Details of the experimental setups can be found in Appendix D.

The results, including AUROC and PR-AUC scores, are presented in Table 4. The performance of the algorithms demonstrates that HINT (Fu et al., 2022b) outperforms the baselines by a significant margin. This is attributed to HINT's incorporation of multi-modal trial components such as molecule structures and disease ontology. Building upon HINT, SPOT (Wang et al., 2023c) further enhances the predictions by employing a sequential modeling strategy for trial outcomes. Notably, AnyPredict (Wang et al., 2023a) achieves the best performance by transfer learning leveraging the data from the Trial Termination Prediction dataset.

## 3.3 TRIAL SEARCH

Table 5: The benchmarking results for *trial search* for the Trial Similarity dataset. Results are precision@K (prec@K) and recall@K (rec@K), and nDCG@K for trial similarities (ranking). The best are in bold.

| Method | Prec@1 | Prec@2 | Prec@5 | Rec@1 | Rec@2 | Rec@5 | nDCG@5 |
|---|---|---|---|---|---|---|---|
| BM25 | 0.7015 | 0.5640 | 0.4246 | 0.3358 | 0.4841 | 0.7666 | 0.7312 |
| Doc2Vec | 0.7492 | 0.6476 | 0.4712 | 0.3008 | 0.4929 | 0.7939 | 0.7712 |
| WhitenBERT | 0.7476 | 0.6630 | 0.4525 | 0.3672 | 0.5832 | 0.8355 | 0.8129 |
| Trial2Vec | **0.8810** | **0.7912** | **0.5055** | **0.4216** | **0.6465** | **0.8919** | **0.8825** |

We conducted an evaluation of the trial search models on the trial search dataset released in (Wang & Sun, 2022c). We have involved the classic probabilistic retrieval algorithm BM25 (Trotman et al., 2014), and the distributional document embedding model Doc2Vec (Mikolov et al., 2013) as the baselines. We also incorporate the sentence transformers based on pre-trained language models WhitenBERT (Huang et al., 2021a) as the comparison to the recent dense trial search algorithm Trial2Vec (Wang & Sun, 2022c). The details of the experimental setups can be found in Appendix E.

The ranking performances are presented in Table 5. The results indicate that the plain BERT model (WhitenBERT) (Huang et al., 2021a) only provides a slight improvement compared to traditional retrieval algorithms like BM25 (Trotman et al., 2014). However, Trial2Vec (Wang & Sun, 2022c), which considers the meta-structure of clinical trial documents and employs hierarchical trial encoding, achieves superior retrieval results.

## 3.4 TRIAL PATIENT SIMULATION

We select a series of synthetic patient data generation algorithms from the electronic healthcare records (EHRs) generation literature, including EVA (Biswal et al., 2021), SynTEG (Zhang et al., 2021), and PromptEHR (Wang & Sun, 2022a). We report their performance to emphasize the importance of developing trial-specific patient simulation models. We also evaluate a recent trial patient generation model Simulants (Beigi et al., 2022) and a personalized trial patient generation model TWIN (Das et al., 2023). We report the fidelity of the generated synthetic patient data based on the

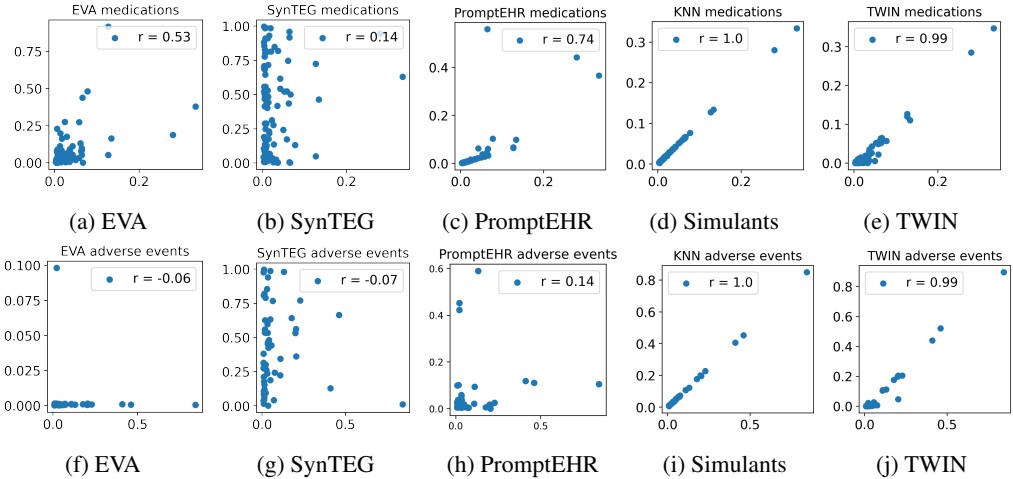

Figure 2: The benchmarking results for *trial patient simulation* on the sequential patient data. Results are dimension-wise probabilities for medications and adverse events (fidelity evaluation). The $x$- and $y$-axis show the dimension-wise probability for real and synthetic data, respectively. $r$ is the Pearson correlation coefficient between them; a higher $r$ value indicates better performance.

sequential trial patient data, which were released in (Das et al., 2023), in Figure 2. The descriptions of these algorithms and experimental setups are available in Appendix F.

In Figure 2, $r$ indicates the affinity of the synthetic data with the real data. We find deep neural networks, which were originally proposed for EHRs, such as EVA (Biswal et al., 2021), SynTEG (Zhang et al., 2021), and PromptEHR (Wang & Sun, 2022a) struggled to fit the data due to the limited sample size. In contrast, Simulants (Beigi et al., 2022) builds on a perturbation strategy with KNN while it produces data that closely resembles real data. Similarly, TWIN (Das et al., 2023) is equipped with a carefully designed perturbation approach by variational auto-encoders (VAE) (Kingma & Welling, 2013) to maintain high fidelity while boosting privacy.

## 4 CONCLUSION

This paper presents `PyTrial`, an innovative Python package designed for advancing research in ML-driven clinical trial development. The package offers a comprehensive suite of 34 machine learning methods customized for addressing six prevalent clinical trial tasks, alongside a collection of 23 readily available ML datasets. `PyTrial` establishes an intuitive and adaptable framework, complete with illustrative examples demonstrating method application and simplified workflows, enabling users to achieve their objectives with just a few lines of code. The package introduces method standardization through its four distinct modules, aiming to simplify and expedite ML research in drug development. This empowers researchers to explore a wide array of challenges in clinical trials using an extensive array of ML techniques. We further discuss the ethical impact of `PyTrial` in Section A.

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

**Contents**

## A  ETHICS & SOCIETAL IMPACT

Though AI shows great promise in boosting drug development, there are still significant ethics and societal impacts that need to be discussed and addressed. We categorize the potential issues into:

**Algorithmic Bias** Machine learning (ML) algorithms are usually optimized by minimizing the average empirical risk. It may raise a concern about the trained algorithms being biased towards the major populations in the training data while undermining the utility for the minority (Mehrabi et al., 2021). We break down the discussion into *predictive* and *generative* algorithms in our case:

- *Predictive model* may perform badly for long-tailed examples. For instance, biased patient outcome prediction or recruitment models may lead to selection bias among clinical trial participants. As such, the in vivo experiment results may not reveal a comprehensive view of effectiveness and adverse effects on a broad population. For this reason, we declare that predictive models should not be used for the participant selection process. Also, in the future, we will support adding algorithmic fairness regularization to predictive models (Kamishima et al., 2011).

- *Generative model* may tend to rehearse only the frequent patterns learned from the training populations, hence decreasing the diversity of the generated populations. The analysis or algorithms developed on the generated data may then inherit this bias. To mitigate this bias, our software develops a recent generation algorithm TWIN (Das et al., 2023) that can produce personalized patient digital twin generation. As such, users can balance the generated synthetic cohort by augmenting for the minority groups. We expect to add more of these types of algorithms in the future.

**Data Privacy**. Our package offers a solution to run ML4Trial algorithms in the local environment, so it does not raise privacy concerns for most use cases. Nonetheless, releasing them to the other party may still cause potential patient data leakage when the generated synthetic records are not being fully audited. To avoid it, our package offers a suite of evaluation functions to audit the results of the generated synthetic patient data, including the evaluation of privacy risks in multiple aspects.

**Responsible Use of AI**. It is crucial to ensure the responsible use of AI in clinical trials; we list several important topics we consider:

- *Human Oversight*. Despite the power of AI algorithms, they should not replace the judgment and experience of medical professionals because AI models make incorrect predictions or lead to adverse outcomes. Human experts should maintain a central role in the decision-making process and use AI as a source of valuable insights, ensuring the correct interpretation and audit of the AI results.

- *Transparency and Accountability*. The practitioners need to ensure transparency in the AI-driven drug development process. It includes detecting and rectifying biases, errors, or unintended consequences produced by AI algorithms. It also requires the timely disclosure of the use of AI algorithms and the results.

- *Adherence to Ethical Guidelines*. Developing and deploying AI algorithms in clinical trials should adhere to the established ethical guidelines and standards. The use of AI should not compromise patient privacy, safety, or well-being. We modify the license of our software to reflect this principle referring to the template in (Contractor et al., 2022).

In summary, we expect AI can boost clinical trials significantly in the future. Nonetheless, it is vital to align AI practices with ethical principles to minimize the pitfalls and ensure the advancement of ML4Trial in an ethical, equitable, and patient-centric manner.

## B  EVALUATION METRICS

### B.1  PREDICTION

The evaluation metrics for the prediction can be categorized into four types: binary classification (ACC, AUROC, PR-AUC), multi-class classification (ACC), multi-label classification (F1-score, Jaccard score), and regression (MSE).

- *ACC*. It measures the ratio of the predicted class that matches the ground truth label. The range is $[0.0, 1.0]$, where $1.0$ is the best.

- *AUROC*. It is the area under the receiver operating characteristic (ROC) curve where the $x$-axis is the false positive rate (FPR) and the $y$-axis the true positive rate (TPR). For binary classification, AUROC indicates the capability of the model to identify positive samples, e.g., mortality, from the testing datasets. The common range of it is $[0.5, 1.0]$, where $1.0$ is the best.

- *PR-AUC*. It is the area under the precision-recall curve that depicts the precision ($y$-axis) against recall ($x$-axis). The curve is obtained by varying the probability threshold that the classifier uses to predict whether a sample is positive or not. It works as a way to measure binary prediction performances. Compared to AUROC, PR-AUC usually fits better for class imbalance datasets. The common range of it is $[0.0, 1.0]$, where $1.0$ is the best.

- *F1-score*. It is a measure of prediction performance by making a harmonic mean of the precision and recall, i.e.,

$$F_1 = 2 * \frac{\text{precision} * \text{recall}}{\text{precision} + \text{recall}}. \tag{1}$$

  The range is $[0.0, 1.0]$, where $1.0$ is the best.

- *Jaccard score*. It is used to compute the similarity between two asymmetric binary variables so as to fit the multi-label prediction scenario. Consider the predicted label set is $\hat{y} \in \{0, 1\}^C$ where $C$ is the number of classes, and the ground truth is $y \in \{0, 1\}^C$. Jaccard score calculates the overlapping

$$\text{Jaccard}(\hat{y}, y) = \frac{|\hat{y} \cap y|}{|\hat{y} \cup y|}. \tag{2}$$

  The range is $[0.0, 1.0]$, where $1.0$ is the best.

- *MSE*. It measures how close the regression predictions match the ground truth labels by calculating the average squared error on the test set as $\frac{1}{N} \sum (\hat{y} - y)^2$. The range is $[0.0, \infty)$, where $0.0$ is the best.

## B.2 RANKING

The evaluation metrics for the ranking task are precision@K, recall@K, and nDCG@K.

- *precision@K*. It counts how precise the top-$K$ results are, as

$$\text{precision@k} = \frac{\text{\# of relevant cases in the top k results}}{k}. \tag{3}$$

  The range is $[0.0, 1.0]$, the best is $1.0$.

- *recall@K*. It counts how good the model captures all relevant cases from the candidate set, as

$$\text{recall@k} = \frac{\text{\# of relevant cases in the top k results}}{\text{\# of relevant trials in all candidate}}. \tag{4}$$

  The range is $[0.0, 1.0]$, the best is $1.0$.

- *nDCG@K*. The full name is normalized discounted cumulative gain (nDCG). It considers the ranking performance in the retrieved set, as

$$\text{nDCG}_k = \frac{\text{DCG}_k}{\text{IDCG}_k}, \tag{5}$$

  where we have

$$\text{DCG}_k = \sum_{i=1}^{k} \frac{\text{rel}_i}{\log_2(i+1)}, \tag{6}$$

$$\text{IDCG}_k = \text{maximize DCG}_k, \tag{7}$$

  where $\text{rel}_i \in \{0, 1\}$ indicates whether the $i$-th retrieved samples are relevant or not. So the range of nDCG@K is $[0.0, 1.0]$, where $1.0$ is the best.

### B.3 GENERATION

We involve three genres of evaluation metrics for synthetic data generation in terms of *privacy*, *fidelity*, and *utility*.

**Privacy** We evaluate privacy to ensure that the information disclosed in the synthetic data is not sensitive and cannot be traced back to the original data. There are three types of metrics:

- *Presence disclosure.* Presence disclosure refers to the scenario where an attacker, possessing a set of patient records denoted as $\mathcal{X}_q$, seeks to determine if any individuals $\mathbf{X} \in \mathcal{X}_q$ are present in the model's training set $\mathcal{X}$. This attack is known as a *membership inference attack* for ML models. We calculate the risk by using both synthetic data $\hat{\mathcal{X}}$ and compromised evaluation data $\mathcal{X}_q$ as

$$\text{Sensitivity} = \frac{\text{\# of known records discovered from synthetic data}}{\text{total \# of known records}}. \tag{8}$$

  It is in the range $[0.0, 1.0]$, and the lower, the better.

- *Attribute disclosure.* A potential risk is the inference of unknown attributes of a target patient, such as specific medications they are taking, using partial information. This risk arises from synthetic data, as it can provide insights into the distribution of real data, enabling attackers to infer sensitive patient information. The evaluation of this risk involves assessing the extent to which synthetic data enables attribute inference by

$$\text{Mean Sensitivity} = \frac{1}{N} \sum_{v=1}^{N} \frac{\text{\# of unknown features of } v \text{ discovered}}{\text{total \# of unknown features of } v}, \tag{9}$$

  where $v$ is a compromised visit, and $N$ is the total number of compromised visits. It is in the range $[0.0, 1.0]$, and the lower, the better.

- *Nearest neighbor adversarial accuracy risk (NNAA).* NNAA is a privacy loss metric used to quantify the degree to which a generative model exhibits overfitting tendencies on the real dataset. This metric is crucial as overfitting can raise privacy concerns if a method reproduces the training data entirely when generating synthetic data. To calculate NNAA, three datasets of equal size need to be created: the training set $\mathcal{X}$, the synthetic data $\hat{\mathcal{X}}$, and the evaluation data $\mathcal{X}_E$. NNAA is computed using the following formula:

$$\text{NNAA} = \text{dist}(\mathcal{X}_E, \hat{\mathcal{X}}) - \text{dist}(\mathcal{X}, \hat{\mathcal{X}}), \tag{10}$$

  where we have

$$\text{dist}(\mathcal{X}_E, \hat{\mathcal{X}}) = \frac{1}{2} \left( \frac{1}{N} \sum_{i=1}^{N} 1(\delta_{ES}(i) > \delta_{EE}(i)) + \frac{1}{N} \sum_{i=1}^{N} 1(\delta_{SE}(i) > \delta_{SS}(i)) \right), \tag{11}$$

$$\text{dist}(\mathcal{X}, \hat{\mathcal{X}}) = \frac{1}{2} \left( \frac{1}{N} \sum_{i=1}^{N} 1(\delta_{TS}(i) > \delta_{TT}(i)) + \frac{1}{N} \sum_{i=1}^{N} 1(\delta_{ST}(i) > \delta_{SS}(i)) \right). \tag{12}$$

  Here, $\delta_{ES}(i) = \min_j \|x_E^i - x_S^j\|$ is defined as the $\ell_2$-distance between $x_E^i \in \mathcal{X}_E$ and its nearest neighbor in $x_S^j \in \hat{\mathcal{X}}$; $\delta_{TS}(i) = \min_j \|x_T^i - x_S^j\|$ is the distance between $x_S^j$ and $x_T^i \in \mathcal{X}$. Similarly we can define $\delta_{TS}$, $\delta_{ST}$ and $\delta_{SE}$. Here, $\delta_{EE}(i) = \min_{j, j \neq i} \|x_E^i - x_E^j\|$. Similarly, we can define $\delta_{TT}$ and $\delta_{SS}$. $1(\cdot)$ is indicator function.

**Fidelity** We evaluate the fidelity of synthetic data by calculating the *dimension-wise probability* (DP) with the following formula

$$\text{DP}(x) = \frac{\text{\# of visits containing the feature } x}{\text{\# of total visits}}, \tag{13}$$

which is the probability of each feature (e.g., medication events) in the dataset. For the real data $\mathcal{X}$, we can compute a sequence of DPs as $\text{DP}(\mathcal{X}) = \{\text{DP}(x_i)\}_i^n$ where $n$ is the total number of features. Similarly, we obtain DP for synthetic data as $\text{DP}(\hat{\mathcal{X}})$. The overall fidelity score is obtained by plotting the scatter plots of real data DPs v.s. synthetic data DPs for visualization. In addition, we

compute the Pearson correlation $r$ between $DP(\mathcal{X})$ and $DP(\hat{\mathcal{X}})$. $r \in [-1, 1]$ and the best situation is $r = 1$.

**Utility** Quantifying the utility of synthetic data generated for downstream tasks is crucial. One such task is training a machine learning (ML) model to predict the incidence rate of a specific clinical endpoint for patients using the generated data $\hat{\mathcal{X}}$. To create a training dataset from $\hat{\mathcal{X}}$, we define $\mathcal{I}$ as the set of patient record inputs $\hat{X}_i$ and $\mathcal{Y}$ as the corresponding target labels representing the endpoints $\hat{y}_i$. Thus, the utility of $\hat{\mathcal{X}}$ can be defined as:

$$\text{Utility}(\hat{\mathcal{X}}) = V(\mathcal{I}, \mathcal{Y}; \mathcal{A}), \tag{14}$$

where $\mathcal{A}$ is any arbitrary predictive ML algorithm that learns from the dataset $\{\mathcal{I}, \mathcal{Y}\}$. The function $V(\cdot)$ measures the prediction performance of $\mathcal{A}$ after training, typically using metrics such as AUROC. It's important to note that the range of values for $V$ may vary depending on the chosen evaluation metric.

## C    BENCHMARK: PATIENT OUTCOME PREDICTION

**Dataset** Our dataset consists of tabular patient data, which is summarized in Table 6. This data was collected from seven separate oncology clinical trials [2], and each trial has its own unique schema. The dataset comprises distinct groups of patients with varying conditions. Our goal is to train a model that can accurately predict the morbidity of patients, which involves a binary classification task.

Table 6: The statistics of Patient Outcome Prediction Datasets. # is short for the number of. Categorical, Binary, and Numerical show the number of columns belonging to these types. N/A means no label is available for the target task.

| Trial ID | Trial Name | # Patients | Categorical | Binary | Numerical | Positive Ratio |
|---|---|---|---|---|---|---|
| NCT00041119 | Breast Cancer 1 | 3,871 | 5 | 8 | 2 | 0.07 |
| NCT00174655 | Breast Cancer 2 | 994 | 3 | 31 | 15 | 0.02 |
| NCT00312208 | Breast Cancer 3 | 1,651 | 5 | 12 | 6 | 0.19 |
| NCT00079274 | Colorectal Cancer | 2,968 | 5 | 8 | 3 | 0.12 |
| NCT00003299 | Lung Cancer 1 | 587 | 2 | 11 | 4 | 0.94 |
| NCT00694382 | Lung Cancer 2 | 1,604 | 1 | 29 | 11 | 0.45 |
| NCT03041311 | Lung Cancer 3 | 53 | 2 | 11 | 13 | 0.64 |
| External Patient Database | | | | | | |
| MIMIC-IV | | 143,018 | 2 | 1 | 1 | N/A |
| PMC-Patients | | 167,034 | 1 | 1 | 1 | N/A |

**Model implementation**

- *XGBoost* (Chen & Guestrin, 2016): This is a tree ensemble method augmented by gradient-boosting. We use ordinal encoding for categorical and binary features and standardize numerical features via `scikit-learn` (Pedregosa et al., 2011). We encode textual features, e.g., patient notes, via a pre-trained BioBERT (Lee et al., 2020) model. The encoded embeddings are fed to XGBoost as the input. We tune the model using the hyperparameters: *max_depth* in $\{4, 6, 8\}$; *n_estimator* in $\{50, 100, 200\}$; *learning_rate* in $\{0.1, 0.2\}$; We take early-stopping with patience of 5 rounds.
- *Multilayer Perceptron (MLP)*: This is a simple neural network built with multiple fully-connected layers. The model is with 2 dense layers where each layer has 128 hidden units. We tune the model using the hyperparameters: *learning_rate* in $\{$1e-4,5e-4,1e-3$\}$; *batch_size* in $\{32, 64\}$; We take the max training *epochs* of 10; *weight_decay* of 1e-4.
- *FT-Transformer* (Gorishniy et al., 2021): This is a transformer-based tabular prediction model. The model is with 2 transformer modules where each module has 128 hidden units in the attention

---

[2]https://data.projectdatasphere

layer and 256 hidden units in the feed-forward layer. We use multi-head attention with 8 heads. We tune the model using the hyperparameters: *learning_rate* in {1e-4,5e-4,1e-3}; *batch_size* in {32, 64}; We take the max training *epochs* of 10 and *weight_decay* of 1e-4.

- *TransTab* (Wang & Sun, 2022b): This is a transformer-based tabular prediction model that is able to learn from multiple tabular datasets. Following the transfer learning setup of this method, we take a two-stage training strategy: first, train it on all datasets in the task, then fine-tune it on each dataset and report the evaluation performances. The model is with 2 transformer modules where each module has 128 hidden units in the attention layer and 256 hidden units in the feed-forward layer. We use multi-head attention with 8 heads. We tune the model using the hyperparameters: *learning_rate* in {1e-4,5e-4,1e-3}; *batch_size* in {32, 64}; We take the max training *epochs* of 10 and *weight_decay* of 1e-4.

- *AnyPredict* (Wang et al., 2023a): This is a transformer-based tabular prediction model built on BERT. The model has 12 transformer layers and was initialized using BioBERT checkpoint [3]. It is equipped with GPT-3.5 (Ouyang et al., 2022) via OpenAI's API [4] for data consolidation and enhancement. This model is able to make pseudo annotations and learn from external patient datasets, which are MIMIC-IV EHR and PMC-Patient notes. We tune the model using the hyperparameters: *learning_rate* in {2e-5,5e-5,1e-4}; *batch_size* in {32, 64}; We take the max training *epochs* of 5 and *weight_decay* of 1e-5.

## D    BENCHMARK: TRIAL OUTCOME PREDICTION

**Dataset** We involve two data sources for this task: TOP benchmark (Fu et al., 2022b) and Trial Termination Prediction (Wang et al., 2023a). Both are tabular prediction tasks: the TOP benchmark's target label is the *success* or *failure* of the trial, which indicates whether the trial meets the primary endpoint or not. By contrast, Trial Termination Prediction has the target label of whether the trial would terminate. The data statistics are in Table 7.

Table 7: The statistics of the Clinical Trial Outcome Datasets. # is short for the number of. N/A means no label is available for the target task.

| Dataset | # Trials | # Treatments | # Conditions | # Features | Positive Ratio |
|---|---|---|---|---|---|
| TOP Benchmark Phase I | 1,787 | 2,020 | 1,392 | 6 | 0.56 |
| TOP Benchmark Phase II | 6,102 | 5,610 | 2,824 | 6 | 0.50 |
| TOP Benchmark Phase III | 4,576 | 4,727 | 1,619 | 6 | 0.68 |
| ClinicalTrials.gov Database | | | | | |
| Trial Termination Prediction | 223,613 | 244,617 | 68,697 | 9 | N/A |

**Model implementation**

- *XGBoost* (Chen & Guestrin, 2016): This is a tree ensemble method augmented by gradient-boosting. We follow the setup used in (Fu et al., 2022b).

- *MLP* (Tranchevent et al., 2019): It is a feed-forward neural network, which has 3 fully-connected layers with the dimensions of dim-of-input-feature, 500, and 100, and ReLU activations. We follow the setup used in (Fu et al., 2022b).

- *HINT* (Fu et al., 2022b): It integrates several key components. Firstly, there is a drug molecule encoder utilizing MPNN (Message Passing Neural Network). Secondly, a disease ontology encoder based on GRAM is incorporated. Thirdly, a trial eligibility criteria encoder leveraging BERT is utilized. Additionally, there is a drug molecule pharmacokinetic encoder, and a graph neural network is employed to capture feature interactions. Subsequently, the model feeds the interacted embeddings into a prediction model for accurate outcome predictions. We follow the setup used in (Fu et al., 2022b).

---

[3] https://huggingface.co/dmis-lab/biobert-v1.1
[4] Engine    gpt-3.5-turbo-0301:    https://platform.openai.com/docs/models/gpt-3-5

- *SPOT* (Wang et al., 2023c): The Sequential Predictive Modeling of Clinical Trial Outcome (SPOT) is an innovative approach that follows a sequential process. Initially, it identifies trial topics to cluster the diverse trial data from multiple sources into relevant trial topics. Next, it generates trial embeddings and organizes them based on topic and timestamp, creating structured clinical trial sequences. Treating each trial sequence as an individual task, SPOT employs a meta-learning strategy, enabling the model to adapt to new tasks with minimal updates swiftly. We follow the setup used in (Wang et al., 2023c).

- *AnyPredict* (Wang et al., 2023a): This is a transformer-based tabular prediction model built on BERT. The model has 12 transformer layers and was initialized using the BioBERT checkpoint. It is equipped with GPT-3.5 (Ouyang et al., 2022) via OpenAI's API for data consolidation and enhancement. This model is able to make pseudo annotations and learn from external datasets, which as the trial termination prediction dataset. We tune the model using the hyperparameters: *learning_rate* in {2e-5,5e-5,1e-4}; *batch_size* in {32, 64}; We take the max training *epochs* of 5 and *weight_decay* of 1e-5.

## E  BENCHMARK: TRIAL SEARCH

**Dataset** The raw training dataset is *Clinical Trial Document* with 447,709 clinical trials, out of which we keep 311,485 interventional trials for self-supervised training. The *Trial Similarity* data is utilized to evaluate the ranking performances of all methods. Given each target trial, the dataset provides 10 candidate trials labeled as $\{1, 0\}$ indicating relevant or not. We can calculate precision@k, recall@k, and nDCG@k, referring to Eqs. equation 3, equation 4, equation 5, based on the ranking results.

**Model implementation**

- *BM25* (Trotman et al., 2014): A bag-of-words retrieval method. We used the default parameters referring to the `rank-bm25` package [5].

- *Doc2Vec* (Mikolov et al., 2013): It is a classic dense retrieval method by building distributed word representations by self-supervised learning methods (CBOW). We take an average pooling of word representations in a document for retrieval by cosine distance. We used the default parameters referring to the `gensim` package [6].

- *WhitenBERT* (Huang et al., 2021a): This is a post-processing method that uses anisotropic BERT embeddings to improve semantic search. We take the average of the last and first layers. This method does not have hyperparameters to choose from.

- *Trial2Vec* (Wang & Sun, 2022c): This is a self-supervised method that utilizes BERT as the backbone and makes a hierarchical encoding of clinical trial documents considering their meta-structure, e.g., the sections. We used the same set of hyperparameters referring to the original paper.

## F  BENCHMARK: TRIAL PATIENT SIMULATION

**Dataset** We utilized the patient records from two clinical trials:

- *Phase III breast cancer trial (NCT00174655)*: A total of 2,887 patients were included in this study, and they were randomly allocated to different treatment groups. The purpose of the study was to assess the effectiveness of Docetaxel, given either sequentially or in combination with Doxorubicin, followed by CMF, compared to Doxorubicin alone or in combination with Cyclophosphamide, followed by CMF, as adjuvant treatments for node-positive breast cancer patients.

- *Phase II small cell lung carcinoma trial (NCT01439568)*: This Phase II trial dataset includes data from both the comparator and experimental arms. A total of 90 patients were randomly assigned to the arms to investigate the impact of LY2510924 in combination with Carbo-platin/Etoposide compared to Carboplatin/Etoposide alone in the treatment of extensive-stage Small Cell Lung Carcinoma.

---

[5] https://pypi.org/project/rank-bm25
[6] https://radimrehurek.com/gensim/models/doc2vec.html

These datasets were pre-processed following the instructions in (Das et al., 2023). Below are the statistics of the processed datasets in Table 8.

Table 8: The statistics of used sequential clinical trial patient data for synthetic data generation tasks.

| Patient: NCT00174655 | | Patient: NCT01439568 | |
|---|---|---|---|
| **Item** | **Number** | **Item** | **Number** |
| # of Patients | 971 | # of Patients | 77 |
| # of Visits | 8,292 | # of Visits | 353 |
| Max # of visits per patient | 14 | Max # of visits per patient | 5 |
| Types of treatments | 4 | Types of treatments | 3 |
| Types of medications | 100 | Types of medications | 100 |
| Types of adverse events | 56 | Types of adverse events | 29 |
| # Patients with severe outcome | 122 | # Patients with severe outcome | 56 |

**Model implementation**

- *EVA* (Biswal et al., 2021): this method leverages variational autoencoders (VAE) to fit the sequential EHRs. EVA is equipped with a hierarchically factorized conditional variant of VAE to make controlled generation. We used the default hyperparameters reported in the original paper.

- *SynTEG* (Zhang et al., 2021): this method consists of a recurrent neural network (RNN) and a generative adversarial network (GAN). It utilizes RNN to encode the historical state of patients while using GAN to learn to generate synthetic visits. We used the default hyperparameters reported in the original paper.

- *PromptEHR* (Wang & Sun, 2022a): this method builds on encoder-decoder language models that perform causal language modeling on the serialized EHRs. It is also capable of making conditional generation through prompt learning. We used the default hyperparameters reported in the original paper.

- *Simulants* (Beigi et al., 2022): it is a simple method that perturbs each real patient record by randomly extracting the corresponding pieces from its nearest neighbors. We used the default hyperparameters reported in the original paper.

- *TWIN* (Das et al., 2023): it was proposed to generate personalized clinical trial digital twins that closely resemble each individual's properties. It introduces VAE for generating the perturbations to produce synthetic records from the target record. We used the default hyperparameters reported in the original paper.

**Fidelity evaluation** We performed a fidelity evaluation on a different patient dataset from a sequential trial (Patient: NCT01439568), and the results are presented in Figure 3. It is observed that both EVA and SynTEG exhibit poor performance on this small dataset, which consists of only 77 patients and 353 visits. PromptEHR, however, shows slightly better performance than the former two. On the other hand, TWIN achieves slightly better results compared to Simulants.

**Privacy evaluation**

- *Presence disclosure*: We randomly select $m\%$ real patient records to set the attacker's data $\mathcal{X}_q$, where $m \in \{1\%, 5\%, 10\%, 20\%\}$. Then, we calculate the Sensitivity scores referring to Eq. equation 8. Results are shown in Table 9. We identify that although Simulants produce high-fidelity synthetic data, it encounters high privacy risk because it turns out to replicate the real patient records when sampling and shuffling across the nearest neighbors of the target patient. On the contrary, TWIN is able to produce novel visits and hence faces less privacy risk.

Table 9: The presence disclosure sensitivity scores with a varying ratio of known samples by the attacker. The lower the value, the better.

| # of known samples in $\mathcal{X}_q$ | 1% | 5% | 10% | 20% |
|---|---|---|---|---|
| Simulants | 0.21 | 0.53 | 0.43 | 0.40 |
| TWIN | **0.16** | **0.21** | **0.20** | **0.19** |

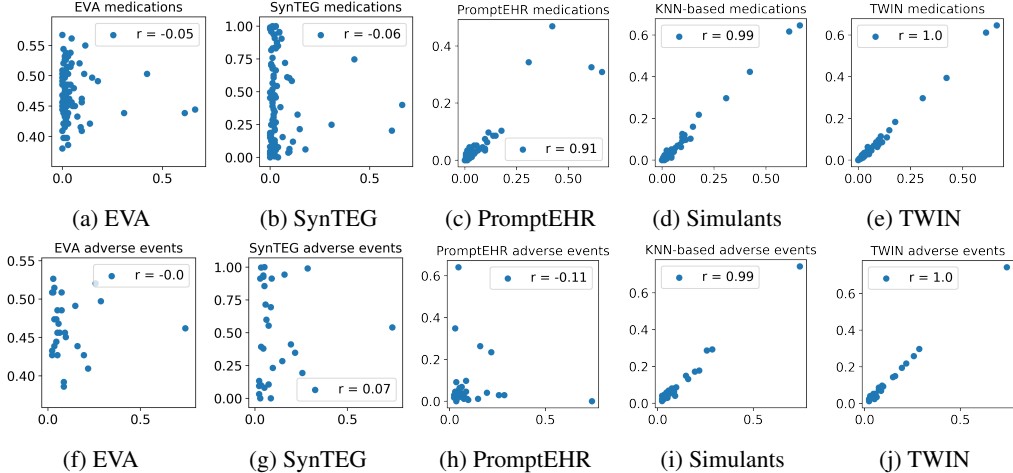

Figure 3: The benchmarking results for *trial patient simulation* on the sequential patient data (NCT01439568). Results are dimension-wise probabilities for medications and adverse events (fidelity evaluation). The $x$- and $y$-axis show the dimension-wise probability for real and synthetic data, respectively. $r$ is the Pearson correlation coefficient between them; a higher $r$ value indicates better performance

- *Attribute disclosure*: We hold out a set of patients and assume that the attacker has access to $m$ of features of these records, where $m \in \{5, 10, 15, 20\}$ in our experiments. We then calculate the Mean Sensitivity scores according to Eq. equation 9. Results are shown in Table 10. It is found that Simulants bear a higher attribute disclosure risk than TWIN in most cases.

Table 10: The attribute disclosure sensitivity scores with a varying number of known features by the attacker. The lower the value, the better.

| # of compromised features | 5 | 10 | 15 | 20 |
|---|---|---|---|---|
| Simulants | **0.221** | 0.283 | 0.303 | 0.394 |
| TWIN | 0.258 | **0.261** | **0.272** | **0.243** |

- *Nearest neighbor adversarial attack risk*: We compute NNAA risk score according to Eq. equation 10, where we hold 100 patients' records from the original real dataset to formulate the evaluation data $\mathcal{X}_E$ and another 100 to formulate the training set $\mathcal{X}$. We observed that TWIN has NNAA score of 0.275 while Simulants has 0.300.

## G  BENCHMARK: TRIAL SITE SELECTION

**Dataset** In these experiments, we utilized real-world clinical trials and claims data as the basis for our research. The dataset consists of 33,323 sites, which were matched with 4,392 trials. Each site was associated with static features, including specialty information, the most recent patient diagnoses and prescriptions, and past enrollment histories. To establish connections between trials and sites, each trial was matched with a fixed number, denoted as $M$, of sites from the available pool.

To introduce the challenge of missing data, we created 10 versions of each trial. For each version, a random mask was generated to determine whether each modality for each site would be present in the data point. While the trial data is publicly accessible, the original site data and enrollment labels are proprietary. Therefore, we started with the real representations of the trials and constructed a synthetic version of the dataset. We accomplished this by creating a pool of 30,000 sites. To generate the static and medical history features for these sites, we randomly sampled from univariate and conditional distributions based on the characteristics of the real dataset. Subsequently, using an enrollment prediction model trained on the real labels, we simulated the enrollment history and

labels for the entire synthetic dataset. Finally, we applied the same augmentation techniques used in the real dataset to introduce the challenge of missing modalities to the synthetic dataset.

**Model implementation** The following methods are included:

- *Random*: selects a subset of sites at random.
- *One-Sides Policy Gradient (PGOS)* (Singh & Joachims, 2019): it is a fairness baseline that adopts a regularization approach to mitigate the underrepresentation of demographic groups within rankings. Rather than explicitly optimizing diversity, this approach focuses on constraining or regularizing fairness to ensure that all demographic groups are adequately represented in the rankings. By incorporating regularization techniques, the baseline model aims to minimize biases and promote equitable outcomes by preventing the systematic underrepresentation of any particular group.
- *Doctor2Vec* (Biswal et al., 2020): it utilizes static features and patient visits to construct a doctor representation. This representation is subsequently queried by a trial representation and passed through a downstream network to predict the doctor's enrollment count for that particular trial. Therefore, it is trained on the smaller, full-data version of the dataset prior to repeating each trial ten times to account for the challenge of missing modalities.
- *FRAMM* (Theodorou et al., 2023a): it is a deep reinforcement learning framework proposed for optimizing site selection for clinical trials. It can handle missing modalities with a modality encoder. It also seeks to reach a trade-off between the enrollment and fairness metrics through specific reward functions. It leverages deep Q-learning to approximate the contribution of each individual site.

**Metrics** We include the following metrics for *enrollment* and *diversity*, respectively.

- *Enrollment*: we compare the size of each model's enrolled cohort with the ground truth maximal enrollment via relative error, calculated by

$$\text{Relative Error} = \frac{\text{Max Enrollment} - \text{Model Enrollment}}{\text{Max Enrollment}} \quad (15)$$

- *Diversity*: we use the entropy of the overall racial distribution of the final enrolled population, defined by

$$H(\mathbf{p}) = -\sum_{k=1}^{6} p_k \log p_k \quad (16)$$

**Experimental results** Our findings are presented from two perspectives: the ability to enroll large patient populations and the simultaneous consideration of diversity within that cohort. We evaluate the performance in terms of enrolling a substantial number of patients for a given study while also ensuring a balanced representation of diverse individuals within the enrolled population. By addressing both aspects, we aim to provide a comprehensive assessment of the effectiveness of the approach in achieving enrollment goals while promoting inclusivity and diversity.

- *Enrollment evaluation*: To evaluate the models' performance in terms of enrollment-focused site selection, we present the enrollment-only results in Table 11. FRAMM achieves the highest enrollment performance on the full-data test set, despite being trained on a distinct data type. Similar results are found on the synthetic dataset where FRAMM reduces relative error up to 74% on the missing data test set as compared to the Random baseline shown in Table 12.

Table 11: Enrollment-Only Performance

|  | Relative Error ($\downarrow$) |
| --- | --- |
| Random | $0.621 \pm 0.019$ |
| Doctor2Vec | $0.525 \pm 0.021$ |
| FRAMM | $\mathbf{0.512 \pm 0.020}$ |

Table 12: Synthetic Enrollment-Only Performance

|  | Relative Error |
| --- | --- |
| Random | $0.227 \pm 0.003$ |
| FRAMM | $0.062 \pm 0.001$ |

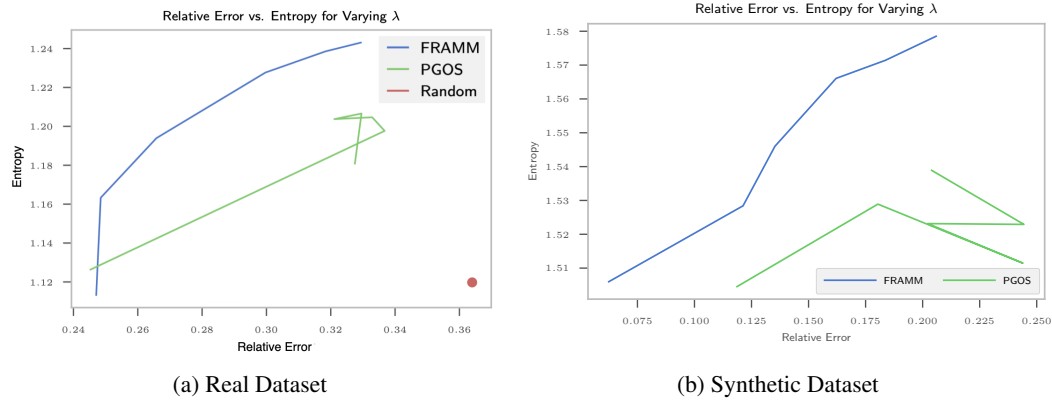

(a) Real Dataset                    (b) Synthetic Dataset

Figure 4: Enrollment vs. Diversity Tradeoffs

- *Diversity evaluation*: In Figure 4a, we illustrate the ability of each method to strike a balance between enrollment and diversity by plotting the trajectories of relative error against entropy while varying the weightings of the fairness component in the loss function. We compare FRAMM to the PGOS and Random baselines using the real dataset. The results demonstrate that both FRAMM and the PGOS model outperform the Random baseline in enhancing diversity. However, FRAMM exhibits superior performance by enabling more efficient and adjustable trade-offs between enrollment and diversity compared to PGOS. It maintains higher enrollment rates for a given level of diversity and offers precise control over the diversity level through different weightings ($\lambda$), whereas PGOS is generally limited to a specific region once $\lambda$ is increased from 0. Similar patterns emerge in the synthetic dataset, as depicted in Figure 4b, where FRAMM continues to demonstrate its capability in achieving more efficient and customizable trade-offs between enrollment and diversity compared to the PGOS baseline.

