# OpenReview forum: "PyTrial: Machine Learning Software and Benchmark for Clinical Trial Applications"
_ICLR.cc/2024/Conference — ICLR 2024 Conference Withdrawn Submission_

### Official Review · Reviewer_dwRR · 2023-10-31

**Soundness:** 2 fair
**Presentation:** 2 fair
**Contribution:** 2 fair
**Rating:** 5
**Confidence:** 3

**Summary:**

The authors propose an ML benchmark for tasks relative to clinical trials.
They define six different related tasks: Patient Outcome Prediction, Patient-Trial Matching, Trial Site Selection, Trial Search, Trial Outcome Prediction, and Patient Data Simulation.
For each task, they implement up to six existing methods in the field and related metrics.

**Strengths:**

This work is particularly significant because of its relevancy to the field and its scale.

### Novelty
As for other medical fields, the lack of reproducible benchmarks allowing a fair comparison between methods hurts the progress of ML in the field making this work highly relevant. I'm not an expert in the field of ML4Trial, but to the best of my knowledge, this work is the first to propose a benchmark of the genre, making it novel.

### Scale
Whether it is in terms of the number of tasks, data sources or baseline implemented, according to the manuscript, this is a huge body of work that has to be acknowledged.

**Weaknesses:**

If there has noticeably been a lot put into this benchmark paper, some fundamental aspects for a good benchmark are missing, strongly weakening the contribution of the authors.

### No source code
First and foremost, the authors did not provide any code. One core need for people in a field to use a benchmark is a smooth, bug-free, implementation. Without this, any benchmark contribution is significantly diminished, as it forbids any guarantee in terms of reproducibility for further use.

### No discussion of data processing and implementation
Assuming there is some existing implementation of the benchmark. As mentioned it would be relevant if there is a normalized procedure for data processing and splitting to ensure a fair comparison between contiguous works in time. Because the code is missing it's again impossible to verify. There is no discussion of such in the manuscript. Again, if the authors leave it to the users to define their own splits and processing pipeline, then their contribution to the community is greatly diminished.

### No analysis of the results

Finally, if we put aside the interest for further use and reproducibility, a core aspect of benchmarking different methods, is to provide further analysis than ranking considered methods. Typically on the underlying reason for certain methods' superiority/inferiority or gives, or analyses regions in which each method performs better. Unfortunately, there is no further analysis of the reason behind the benchmark results.

### Conclusion
There is a clear need for an ML4Trial benchmark and the authors report a great amount of work concerning that effort by defining numerous tasks and implementing many existing baselines.  However, the lack of guarantee in terms of usage and reproducibility as well as the shallowness of the results analysis, significantly decrease the authors' contribution.

**Questions:**

I don't have specific questions beyond the points I raised in the weaknesses.

---

### Official Review · Reviewer_cWL5 · 2023-10-31

**Soundness:** 2 fair
**Presentation:** 3 good
**Contribution:** 3 good
**Rating:** 5
**Confidence:** 5

**Summary:**

The work proposes a library for benchmarking various tasks related to clinical trials. The work surveys many tasks and approaches from literature and organizes them into different sections of the library. The library seems to be a sklearn like system focused on pandas and numpy.

**Strengths:**

The great benefit of a library like this is the ability to perform side by side analysis easily on the same data configuration across multiple methods. This library can likely serve as the framework that new methods are implemented in.

**Weaknesses:**

It was difficult for me to locate the demo notebooks. I would be better to store them in the repo and then link to them using a colab link like this: https://colab.research.google.com/github/RyanWangZf/PyTrial/notebooks/demo.ipynb and you can filter this folder out from the pip package so it won't be bundled in there.

I think it would be helpful if you prepared benchmark subsets of data along with the demo data so users wouldn't have to compile their own data. Then they could focus more on the algorithm design. Right now I see a file called "phase_I_train.csv" and I have no way to reference this in a paper so that others can also use the same file. It could be updated at any time.


I tried to load some of the "23 readily available ML datasets" referenced by the paper but I was unable to find documentation on how to do this. The data package 'data.trial_data' contains two classes that take pandas dataframes as input and it is unclear how they should be structured. In the tutorial section 'Individual Patient Outcome Prediction' appears to only use the demo data and then synthetic data for the sequence models. Improving the documentation about all the datasets and how to access them would greatly improve the usefulness of this library.

Also in order to install the library I needed to remove the version requirement for sdmetrics. I'm using python 3.11. It seemed to work after that. I would have run unit tests to check but there are none present in the library. You should consider adding some.

**Questions:**

Noted in weaknesses.

---

### Official Review · Reviewer_BJYq · 2023-11-01

**Soundness:** 3 good
**Presentation:** 3 good
**Contribution:** 2 fair
**Rating:** 5
**Confidence:** 4

**Summary:**

This paper introduced the a software package called PyTrial. PyTrial integrates more than 20 trial related datasets (23 from from PDS, PubMed, MIMIC, SIGIR, ClinicalTrials.gov, PrimeKG, Drugbank, and WHO in 4 categories: Patient baseline data, Trial key features, drug calcification, and disease clarification. Also include the WHO ATC codes.) More than 30 ML algorithms from existence literates are built into PyTrial (11 for patient outcome prediction, 2 for trial site selection, 6 for trial outcome prediction, 2 for patient-trial matching, 4 for trial search, and 11 for trial patient simulation).
The paper is publicly accessible at https://arxiv.org/abs/2306.04018 and the package can be located under https://github.com/RyanWangZf/PyTrial.

**Strengths:**

This paper introduced a software platform which can load data in specific format (more than 20 existing datasets pre-loaded) and apply build-in 30+ ML-methods easily by loading data, select model and run the simple user interface to fit and predict. The amount of work to build those together can be tremendous. This can be a nice starting platform for exploring trial related ML-related outcome.

**Weaknesses:**

The trial site selection, trial search and patient-trial matching might be good starting point to further develop the proposed PyTrial platform, though these match would be further limited by other operational/contract issues.

The patient outcome prediction, trial outcome prediction and especially the trial patient simulation might require more complex manipulation of collected features and more availability of disease/drug/patient specific data.

**Questions:**

P1. Clarify when first mention ML4Trial what it is and available source. Otherwise reader may search existing python/R/... packages for it.
P1. Provide path to the 23 ipynb to facilitate reading if you do want to mention it at the beginning.
P2. No impact to the content of this platform, but just curious your source of clinic trial pass rate by phase. Your cited values look high.
P2-3, Table 1 and Table 2 listed datasets and methods are 24 and 36, might be nice if you distinguish which are dictionary/duplicate to match your cited number 23 and 34.
P6. The description of the motivation and impact of trial patient simulation might be future ideal case considering patient data privacy and high bar of regulatory approval of trials.
I tried your software but didn’t look into the simulation results because of limited time and your paper is more about introducing the platform while the methods are from existing papers.

---

### Official Review · Reviewer_nm57 · 2023-11-01

**Soundness:** 3 good
**Presentation:** 3 good
**Contribution:** 3 good
**Rating:** 5
**Confidence:** 3

**Summary:**

This paper introduces a comprehensive benchmarking tool called PyTrial, designed to integrate mainstream machine learning methods for clinical trial tasks. PyTrial encompasses six machine learning tasks related to clinical trials, including patient outcome prediction, patient-trial matching, trial site selection, trial search, trial outcome prediction, and patient data simulation. Additionally, this tool provides over 20 machine learning-ready datasets for the purpose of validation and development of machine learning models. Furthermore, PyTrial establishes standard evaluation procedures for all tasks, such as accuracy for prediction tasks, precision/recall for ranking tasks, and privacy, fidelity, and utility for generation tasks. For all the algorithms included in PyTrial, practical examples in Jupyter Notebook are provided to ensure ease of testing and implementation. In summary, PyTrial offers a comprehensive interface to facilitate the rapid implementation of ML4Trial algorithms on users' own data and the deployment of these algorithms to enhance clinical trial planning and execution.

**Strengths:**

The PyTrial tool provided in this paper offers a complete workflow with several notable advantages:

1. Open-Source and Well-Documented: The code is open-source and comes with comprehensive documentation, ensuring transparency and ease of use for researchers and practitioners.

2. Simplified Code Framework: The tool's code framework is concise and user-friendly, featuring a standardized workflow, making it accessible even for those with limited experience in machine learning.

3. Incorporation of Benchmark Datasets: PyTrial includes curated benchmark datasets for various tasks, facilitating meaningful comparisons and evaluations in different contexts.

4. Integration of Model Methods: Within the tool, common baseline model methods for different tasks are integrated. Additionally, it provides a range of both classic and cutting-edge model methods specific to different tasks, allowing for thorough comparative assessments.

In summary, PyTrial not only streamlines the implementation of ML4Trial algorithms on users' own data but also enhances reproducibility and benchmarking in the field of clinical trials, offering a valuable resource for researchers and professionals.

**Weaknesses:**

The author just show the benchmark analysis in the paper, lacks the case study on how to expand the framework or incoorperate new tasks or algorithms. In addition, the most of the integrated datasets are Tabular/Sequential, it seems that some parts of the original datasets are abandoned (e.g. raw text)

Despite the authors' efforts in providing a Python toolkit and developing comprehensive data-model-evaluation workflows for specific tasks, there are concerns regarding the potential limited reach and impact of this tool within the field. From the perspective of providing benchmark pipeline alone, this work does not appear to offer a significant advantage or possess substantial influence. Furthermore, the tool's utility may require extensive real-world applications and evaluations by relevant domain researchers to gain recognition within the community. It is not advisable to present the software in the form of a paper before it has been widely adopted and utilized.

**Questions:**

1. Advantages Over HuggingFace Framework: Can you please clarify the advantages of this software over existing frameworks like HuggingFace? Additionally, have you considered the possibility of uploading this work to HuggingFace to enhance its impact and accessibility?

2. Existing Competitions Using Text and Tabular Data: Are there any significant competitions that primarily rely on text and tabular data in the field? If so, what are the specific advantages of this tool when compared to openly available competition datasets and evaluations?

3. Automated Data Collection and True-Blind Testing Sets: Is the software capable of implementing automated weekly data collection and cleaning for the latest data? Does it have the capacity to provide regularly updated true-blind independent test sets to improve the scale and quality of benchmark data?

4. Data usage agreement:  Datasets like MIMIC-III and MIMIC-IV require the user to sign the agreement. How to address the problem for the user of PyTrial?